# Gold Nanoparticles Enhance the Tumor Growth-Suppressing Effects of Cetuximab and Radiotherapy in Head and Neck Cancer In Vitro and In Vivo

**DOI:** 10.3390/cancers15235697

**Published:** 2023-12-03

**Authors:** Takumi Sato, Yasumasa Kakei, Takumi Hasegawa, Masahiko Kashin, Shun Teraoka, Akinobu Yamaguchi, Ryohei Sasaki, Masaya Akashi

**Affiliations:** 1Department of Oral and Maxillofacial Surgery, Kobe University Graduate School of Medicine, 7-5-2 Kusunoki-cho, Chuo-ku, Kobe 650-0017, Japan; tupac.takusato0225@gmail.com (T.S.); mshkshinma@gmail.com (M.K.); ts25512@gmail.com (S.T.); akashim@med.kobe-u.ac.jp (M.A.); 2Laboratory of Advanced Science and Technology for Industry, University of Hyogo, Kamigori 678-1205, Japan; y258a012@guh.u-hyogo.ac.jp; 3Division of Radiation Oncology, Kobe University Graduate School of Medicine, 7-5-2 Kusunoki-cho, Chuo-ku, Kobe 650-0017, Japan; rsasaki@med.kobe-u.ac.jp

**Keywords:** oral squamous cell carcinoma, irradiation, cetuximab, gold nanoparticle

## Abstract

**Simple Summary:**

Head and neck squamous cell carcinoma (HNSCC) treatment includes surgery, radiotherapy, and immunotherapy with the aim of eradicating cancer cells without affecting normal tissues. HNSCC expresses epidermal growth factor receptor and cetuximab, an IgG1 monoclonal antibody targeting epidermal growth factor receptor, has been approved for the treatment of HNSCC. However, cetuximab has low reactivity. Gold nanoparticles (AuNPs) were reported to enhance the local antitumor effects of radiotherapy without damaging normal cells. This study investigated the in vitro and in vivo effects of single and combination therapy with AuNPs, cetuximab, and radiotherapy on a human HNSCC cell line. Combination treatment of AuNPs + cetuximab + radiotherapy markedly reduced HNSCC cell numbers and proliferation and enhanced apoptosis compared with single and double combination treatments. Furthermore, the in vivo combination treatment (AuNPs + cetuximab + radiotherapy) reduced the tumor volume compared with the controls. This study showed that AuNPs sensitize tumors to radiotherapy and bind to cetuximab, leading to enhanced antitumor effects.

**Abstract:**

Introduction: Head and neck squamous cell carcinoma (HNSCC) treatment includes surgery, radiotherapy, and immunotherapy with the aim of eradicating cancer cells without affecting normal tissues. HNSCC expresses epidermal growth factor receptor (EGFR) and cetuximab, an IgG1 monoclonal antibody targeting epidermal growth factor receptor, has been approved for the treatment of HNSCC. However, cetuximab has low reactivity and induces serious side effects. Gold nanoparticles (AuNPs) were reported to enhance the local antitumor effects of radiotherapy without damaging normal cells. Methods and Results: This study investigated the in vitro effects of single and combination therapy with AuNPs (1.0 nM), cetuximab (30 nM), and radiotherapy (4 Gy) on a human HNSCC cell line, HSC-3. Combination treatment of AuNPs + cetuximab + radiotherapy markedly reduced HSC-3 numbers and proliferation and enhanced apoptosis compared with single and double combination treatments. Furthermore, the in vivo combination treatment (AuNPs + cetuximab + radiotherapy) of a xenograft model of HSC-3 cells transplanted into nude mice (BALB/cAJcl-nu/nu) reduced the tumor volume compared with the controls. Scanning electron microscopy demonstrated the presence of AuNPs in tumor tissues and toxicity analysis indicated that AuNPs had no toxic effect on normal tissues. Conclusions: This study showed that AuNPs alone do not have a tumor-suppressing effect, but they sensitize tumors to radiotherapy and bind to cetuximab, leading to enhanced antitumor effects.

## 1. Introduction

Head and neck squamous cell carcinoma (HNSCC) comprises approximately 1% to 5% of all cancers, and oral cancer is the most common HNSCC [1,2,3,4]. HNSCC most commonly arises in middle-aged and elderly men, and its morbidity is increasing with population aging [5]. HNSCC is treated by surgery, radiotherapy, and immunotherapy, and chemotherapy and radiotherapy can be used in combination after surgery. The goal of treatment is to eradicate cancer cells without affecting normal tissue. HNSCCs express epidermal growth factor receptor (EGFR) [6], which is widely distributed in the tumor surface layer, and increased EGFR expression is often associated with poor prognoses in patients with human epithelial carcinoma [7]. Therefore, EGFR is both a prognostic biomarker and a promising therapeutic target [8]. Cetuximab was the world’s first IgG1 monoclonal antibody targeting EGFR introduced for the treatment of HNSCC [9]. Cetuximab was launched for the treatment of HNSCC based on the results of two important clinical trials. The first study was the Bonner trial, which included patients with locally advanced HNSCC. The median duration of locoregional control was 24.4 months among patients treated with cetuximab plus radiotherapy, compared with 14.9 months among those given radiotherapy alone (hazard ratio for locoregional progression or death = 0.68; *p* = 0.005). After a median follow-up of 54.0 months, median overall survival was 49.0 months among patients treated with the combined therapy versus 29.3 months among those treated with radiotherapy alone (hazard ratio for death = 0.74; *p* = 0.03) [10,11,12]. The second study was the EXTREME study, which enrolled patients with recurrent or metastatic HNSCC. Compared with the group that received the previous standard of care of two-drug chemotherapy (platinum-based therapy and 5-FU), the group that received cetuximab exhibited a 20% reduction in the risk of disease progression or death and a significant increase in median survival from 7.4 to 10.1 months. [13]. However, EGFR inhibitors, including cetuximab, have serious side effects and low reactivity, and the existence of residual tumors after treatment remains an issue to be resolved. Meanwhile, radiotherapy should minimize the impact on normal tissues [10]. Therefore, gold nanoparticles (AuNPs) have attracted attention because they only enhance the local action of radiation. AuNPs have found wide application in biology and engineering because of their characteristic optical properties [14,15], which are attributable to the interaction of electrons and light on the particles. As a result of their high atomic number, gold nanoparticles are ideal radiosensitizers that absorb photons and emit secondary photoelectrons, which may enhance the cell-killing properties of radiation through DNA damage [16]. At certain wavelengths of light, a phenomenon called surface plasmon resonance occurs. In this process, electrons on the particle surface oscillate together, resulting in strong absorption and scattering of light [17]. In this experiment, cetuximab was used instead of the selective EGFR tyrosine kinase inhibitor AG1478 [18], which was used in previous experiments. Gold nanoparticles are attached to inhibitors, such as EGFR, as in our previous study [19]. We propose that if an EGFR inhibitor with gold nanoparticles on its surface is attached to cancer cells, the radiosensitizing effect of the EGFR inhibitor may be enhanced. 

In this study, we hypothesized that the tumor growth inhibitory effect of cetuximab, which is more selective than AG1478, plus radiotherapy would be enhanced by the addition of AuNPs.

## 2. Materials and Methods

(1)Cell Culture

We used the human HNSCC cell line HSC-3 (tongue cancer; provided by the Japanese Collection of Research Bioresource Cell Bank, Tokyo, Japan). HSC-3 cells were cultured in Dulbecco’s modified eagle’s medium (DMEM; Wako Pure Chemical Industries, Ltd., Osaka, Japan) supplemented with 10% (*v*/*v*) fetal bovine serum (Biowest, Nuaillé, France) and 1% penicillin/streptomycin (Sigma-Aldrich; Merck KGaA, Darmstadt, Germany) at 37 °C in a humidified atmosphere of 5% (*v*/*v*) CO_2_. Cells were incubated in DMEM for 48 h prior to treatment. We used cells from a relatively early stage (passages 2–10).

(2)Animal Species

Male nude mice (BALB/cAJcl-nu/nu, 25–27 g, 4–6 weeks old) were purchased from CLEA (Fuji Breeding Facility, Inc., Tokyo, Japan). The nude mice were maintained in specific pathogen-free animal care facilities at 21–25 °C and 40–70% humidity with free access to food and water [20]. All animal experiments were approved by the Kobe University Institutional Animal Care and Use Committee (animal testing approval number P-201207) and performed in accordance with the Kobe University Animal Experimentation Regulations.

(3)AuNPs, Cetuximab, and X-ray Radiation

Citrate-stabilized AuNPs were purchased from Cytodiagnostics, Inc. (Burlington, ON, Canada). Standard AuNPs (60 nm) were used (lot number 2458052_60). Lysosomal uptake of AuNPs was confirmed by transmission electron microscope (JEM-1400Plus; JEOL Ltd., Tokyo, Japan) imaging at an acceleration voltage of 100 kV. Digital images (3296 × 2472 pixels) were taken with a CCD camera (EM-14830RUBY2; JEOL Ltd.) as reported previously [18] (Appendix A). Cells were grown in 2 mL of DMEM on glass slips in 35 × 10 mm^2^ polystyrene tissue culture dishes at a density of 1 × 10^5^ cells/mL. Cells were incubated with AuNPs diluted with 1.0 nM phosphate-buffered saline (PBS) and cetuximab at 0 or 30 nM. After 24 h at 37 °C, samples were irradiated by an MBR-1505R2 X-ray generator (Hitachi, Tokyo, Japan) at 150 kV and 4 mA using a 1 mm aluminum filter. Cells were exposed to a fixed dose of 4 Gy, as reported previously [21]. Subsequently, the experiments were started after incubation at 37 °C for 48 h.

(4)X-ray Radiation In Vivo

Prior to each experiment, the mice were anesthetized using 2% isoflurane in O_2_ and taped on a base made by combining two tongue depressors. Other body surfaces were covered with lead plates to ensure that only tumors were irradiated.

(5)Xenograft Assay

HSC-3 cells (3.5 × 10^6^–4.0 × 10^6^ cells) were mixed with Matrigel (Basement Membrane Matrix, Corning Inc., Shizuoka, Japan) and injected at a volume of 0.1 mL subcutaneously into the back of each nude mouse. After the tumor volume reached 200–300 mm^3^, mice were assigned to one of eight groups: control (treated with PBS), AuNPs (Product Number 765309, Sigma-Aldrich; Merck KGaA, Darmstadt, Germany) (treated with 10 nm diameter AuNP suspension at a concentration of 15 μg mL^−1^), cetuximab (treated with cetuximab suspension at a concentration of 500 μg mL^−1^), AuNPs + cetuximab (treated with 10 nm diameter AuNP suspension at a concentration of 15 μg mL^−1^ and cetuximab suspension at a concentration of 500 μg mL^−1^), radiotherapy (4 Gy), AuNP + radiotherapy (treated with 10 nm diameter AuNP suspension at a concentration of 15 μg mL^−1^ and 4 Gy or radiation), cetuximab + radiotherapy (treated with cetuximab suspension at a concentration of 500 μg mL^−1^ and 4 Gy of radiation), and AuNP + cetuximab + radiotherapy (treated with 10 nm diameter AuNP suspension at a concentration of 15 μg mL^−1^, cetuximab suspension at a concentration of 500 μg mL^−1^, and 4 Gy of radiation). AuNPs, cetuximab, or a combination of both were adjusted to a total of 150 μL in cell culture medium to the final concentrations described above and then injected directly into the subcutaneous tumor on the backs of nude mice. X-ray irradiation was performed using an MBR-1505R2 X-ray generator (Hitachi, Tokyo, Japan) at a voltage of 150 kV and a current of 5 mA with a 1 mm thick aluminum filter (0.5 Gy min^−1^ at the target). The tumor size, body weight, and health status of all mice were monitored every 2 or 3 days for 49 days post-treatment. The volume conversion formula was long axis (mm) × shortest axis (mm) × shortest axis (mm)/2, as reported by Hassan et al. [20]. On day 49, all mice were sacrificed via isoflurane inhalation, and the tumors were excised with the surrounding skin.

(6)Tumor Volume Reduction

Mice were sacrificed, and the skin on the back was collected with the tumor and its margin. The tissue was stored in a 10% neutral buffered formalin solution and mailed to Kobe Kyodo Pathology Co. Ltd. (Kobe, Japan). Tumors were embedded, thinly sliced, and stained with hematoxylin and eosin.

(7)Effect of AuNPs on Normal Organs

HSC-3 cells (3.5 × 10^6^–4.0 × 10^6^ cells) were mixed with Matrigel and injected at a volume of 0.1 mL subcutaneously into the back of each nude mouse. Animals were then divided into the control (treated with PBS) and AuNP groups (treated with 10 nm diameter AuNP suspension at a concentration of 15 μg mL^−1^). These mice were observed for 49 days, as described in the aforementioned experiments. Then, the mice were sacrificed, and the liver, kidneys, heart, and lungs were removed from each individual. Tissues were fixed with a 10% neutral buffered formalin solution and embedded in paraffin blocks. The sections were stained with hematoxylin and eosin and observed using a BZ-X 700 fluorescence microscope (Keyence Corporation, Osaka, Japan).

(8)Cell Counting

Cells were cultured in a 35 × 10 mm^2^ dish with DMEM and placed in an incubator for 24 h. Cells were then treated with cetuximab (0 or 30 nM) and AuNPs (0 or 1 nM). After 24 h of incubation, cells were irradiated. Forty-eight hours after irradiation, cells were fixed with 1% formaldehyde in PBS for 10 min, treated with 0.2% Triton X-100 in PBS for 5 min, washed with PBS, blocked with 1% bovine serum albumin (BSA; Sigma-Aldrich; Merck KGaA), and incubated at 4 °C overnight. The cover glass with cells was washed three times with PBS and soaked in a primary antibody (rat anti-E-cadherin mAb) diluted 200-fold with 10× Tris-buffered saline (pH 7.4, NIPPON GENE) for 3 h at room temperature. Then, cells were washed three times with PBS, soaked in 4′,6-diamidino-2-phenylindole (DAPI; Thermo Fisher Scientific, Inc., Waltham, MA, USA), and incubated with a secondary antibody (1:500; Cy3^®^) for 30 min. The cover glass was again washed three times with PBS and rinsed with Milli-Q water. Then, the cover glass was embedded in FluorSave™ (Merck KGaA) and observed using a BZ-X 700 fluorescence microscope. Five random fields of view were selected, and DAPI-positive cell nuclei were counted in each field. In addition, E-cadherin-positive intercellular junctions were also randomly counted at five locations (magnification, ×40). This experiment was performed independently three times.

(9)Cell Proliferation Assay

The inhibition of cell proliferation by AuNPs and cetuximab was examined. Specifically, 5 × 10^3^ cells in complete medium were added to each well of a 96-well plate and cultured at 37 °C for 24 h. AuNPs and cetuximab were used at the same concentrations as applied in prior experiments. After incubation for 24 h, cells were irradiated. Forty-eight hours after irradiation, a Cell Counting Kit-8 solution (Dojindo Molecular Technologies, Inc., Kumamoto, Japan) was added (10 μL per well). After 1–4 h, absorbance was measured at 450 nm using the Enspire multimode plate reader.

(10)Apoptosis

To assess apoptosis, cells were grown in DMEM on glass slips in 35 × 10 mm^2^ polystyrene tissue culture dishes at a density of 1 × 10^5^ cells/mL for 24 h. We added cetuximab (0 or 30 nM) and AuNPs (0 or 1 nM) to each dish, which were incubated for 24 h, followed by irradiation. Forty-eight hours after irradiation, cells were fixed with 1% formaldehyde for 10 min. After fixing, cells were washed three times with PBS, dipped in 0.2% Triton X-100, and incubated for 5 min. Then, cells were washed with PBS, blocked with 1% BSA, and incubated at 4 °C overnight. Cells were washed three times with PBS and soaked in a primary antibody (rat anti-E-cadherin mAb) diluted 200-fold with a 10× TBS buffer for 3 h at room temperature. After that, cells were washed three times with PBS, soaked in an anti-caspase-3 antibody (Cell Signaling Technology, Danvers, MA, USA) and DAPI, and incubated with a secondary antibody (1:500; Cy3^®^) for 30 min. The cells were washed three times with PBS and rinsed with Milli-Q water. Then, they were embedded in FluorSave™ and observed using a BZ-X 700 fluorescence microscope. The percentage of apoptotic cells was calculated according to the total number of cells.

(11)Scanning Electron Microscopy (SEM)

In this study, the field emission scanning electron microscopes JSM-7100F (JEOL Ltd.) and JSM-6700F (JEOL Ltd.) were used for the observations. A carbon coating with a thickness of approximately 10 nm was used. The acceleration voltage was set at 15 kV. We used the backscattered electron (BSE) compositional image (COMPO image) mode, which is a reflection electron image illustrating the difference in the average atomic number of the sample, or composition, to search for AuNPs. This technique is often used for preliminary observation for elemental analysis by energy-dispersive X-ray spectrometry because it is relatively easy to obtain the compositional distribution of a sample. Specifically, a COMPO image is obtained by summing the output signals of the secondary electron detector and the BSE detector. Because the output signals from the two detectors are summed, the effect of specimen topography is eliminated, and the contrast attributable to the compositional difference is obtained.

(12)Statistical Analysis

For means and standard deviations, the Tukey–Kramer multiple comparison test was used for analysis. *p* < 0.05 denoted statistical significance.

## 3. Results

Cell Counting Assay

We determined the average counts of HSC-3 cells in the eight different experimental groups. Cell counting assay results showed that the average number of cells in each group was 374.4 for the control, 324.1 for AuNPs, 189.6 for cetuximab, and 176.7 for AuNPs plus cetuximab. When radiation was added to each condition, the average number of cells was 148.6, 122.7, 100.2, and 59.5, respectively. The cell count was significantly reduced in all treatment groups versus the control group (*p* < 0.001), excluding the AuNPs alone group, as presented in Figure 1.

Apoptosis Assay

We assessed apoptosis in HSC-3 cells following treatment with AuNPs, cetuximab, and/or radiotherapy by counting the number of cells positively stained for caspase-3. The average percentages of apoptosis-positive cells for the control, AuNPs, cetuximab, and AuNPs plus cetuximab were 0.4%, 0.3%, 4.3%, and 5.8%, respectively, and increased to 4.1%, 9.1%, 25%, and 31% when radiation was added to each treatment. Significant differences were observed between the control and AuNPs + radiation groups, the control and radiation + cetuximab groups, and the control and radiation + cetuximab + AuNPs groups (*p* < 0.001). Significant differences were also observed between the radiation + cetuximab and radiation + cetuximab + AuNPs groups (*p* = 0.0144). The detailed results are presented in Figure 2.

Proliferation Assay

Proliferation assay results showed that the average cell growth percentages were 102% for the control, 120% for AuNPs, 77% for cetuximab, and 93% for AuNPs plus cetuximab, and 88%, 106%, 51%, and 31% when radiation was added to each condition. Cell proliferation was significantly reduced in the AuNPs + cetuximab + radiotherapy group compared with the control group (*p* = 0.0049). The detailed results are presented in Figure 3.

Tumor Growth Inhibitory Effect In Vivo

The tumor size in mice injected with HSC-3 cells was approximately 140–160 mm^3^ prior to treatment.

As shown in Figure 4a, at POD49, the subcutaneous tumor volume had decreased in the treatment groups versus the control and AuNP-only groups. The average volume of the control group was 169.3 mm^3^, the volume of AuNPs was 170.2 mm^3^, cetuximab was 65.7 mm^3^, and AuNPs plus cetuximab was 62.4 mm^3^. Adding radiation to each group resulted in tumor volumes of 82.6 mm^3^, 60.3 mm^3^, 50.2 mm^3^, and 39.8 mm^3^. On day 49 after treatment, the tumor volume was significantly lower in all treatment groups, excluding the AuNPs group, than in the control group (Figure 4b). In addition, tumor volume was significantly smaller in the AuNPs + cetuximab + radiotherapy group than in the cetuximab + radiotherapy group (*p* = 0.0036). Body weight tended to increase in all animals, and no significant difference was observed among the groups (Figure 4c).

Evaluation of Tumor Cross-sections In Vivo

After mice were sacrificed, the dorsal subcutaneous tumor was excised, along with surrounding normal tissue, and stained with hematoxylin and eosin. Representative weakly expanded images are presented in Figure 5.

Observation of AuNP Uptake into Tumors by SEM

Figure 6a,c,e present the normal SEM observation images. The corresponding BSE COMPO images are presented in Figure 6b,d,f. In the COMPO image, the white areas are the locations in which the reflected electron emission rate is high, i.e., where atoms heavier than carbon, the main component of the cell, are present. AuNPs that are hidden by part of the cytoskeleton in Figure 6c,e can be clearly observed in Figure 6d,f.

Toxicity Assessment in Normal Organs In Vivo

To assess the effect of AuNPs on normal organs, the liver, kidneys, heart, and lungs were removed and compared with those in the control group. Tissues were stained with hematoxylin and eosin and observed under a microscope (Figure 7). For each organ, no clear changes were observed in the AuNPs group compared with the findings in the control group. No obvious tissue abnormalities were observed in each organ following AuNP treatment.

## 4. Discussion

This study revealed that combination treatment with AuNPs, cetuximab, and radiotherapy significantly reduced HSC-3 cell counts (Figure 1, Figure 2 and Figure 3). Furthermore, this combination treatment led to tumor shrinkage in vivo, as shown in Figure 4 and Figure 5, without toxic findings in major organs, as shown in Figure 7. As shown in Figure 6, the strength of this study also lies in the demonstration of the localization of gold nanoparticles within the excised tumor. AuNPs alone do not have a tumor-suppressing effect, but they sensitize tumors to radiotherapy and bind to cetuximab, leading to enhanced antitumor effects. (Figure 1, Figure 2, Figure 3, Figure 4 and Figure 5). The reduction in cell counts following treatment with AuNPs, cetuximab, and radiotherapy was attributable to apoptosis (Figure 2). This is the first report to examine the combined effects of AuNPs, cetuximab, and radiotherapy in mice with HNSCC.

Hassan et al. compared the radiosensitizing effects of TiOxNPs and AuNPs in MIA PaCa-2 human pancreatic cancer cells (JCRB0070) and found that TiOxNPs enhanced tumor suppression [20]. This study compared gold nanoparticles with titanium nanoparticles but did not add molecularly targeted drugs, such as cetuximab, which was performed in the current study. In other reports, cetuximab and trastuzumab were used to direct AuNPs toward cancer to enhance the effects of radiotherapy [22,23,24,25,26,27]. Chattopadhyay et al. reported that HER2-targeted AuNPs increased the antitumor effect of radiotherapy in a xenograft model of breast cancer and that AuNPs are not harmful to normal tissue in vitro and in vivo [22,23]. Yook et al. found that AuNPs targeting both HER2 and EGFR had a stronger radiosensitizing effect on breast cancer cell lines than AuNPs targeting each gene alone [24]. The methodology used in these studies was similar to ours but differed in that breast cancer was the subject of the studies. Popovtzer et al. reported that cetuximab targeted with AuNPs for head and neck cancer enhanced the effects of radiotherapy and significantly affected tumor growth, and the mechanism of radiation enhancement was associated with increased apoptosis (TUNEL assay), inhibition of angiogenesis (based on CD34 levels), and decreased repair mechanisms (proliferating cell nuclear antigen staining). Furthermore, they reported that AuNPs were safe because no evidence of toxicity was observed [25]. This study is the only in vivo report using gold nanoparticles, radiation, and cetuximab; however, the cell line used for head and neck cancer was epidermoid carcinoma, a rare histology for human head and neck squamous cell carcinoma, and the authors did not identify the gold nanoparticles in the tumor tissue. The addition of AuNPs in this study increased the antitumor effects of radiotherapy and cetuximab, but the combination did not completely eliminate tumors in vivo. Cetuximab has been clinically applied in combination with taxane-based anticancer agents, such as paclitaxel, in the treatment of difficult-to-resect tumors [26]. Hallal et al. reported that the fixed-dose combination of gold nanoparticles and cetuximab itself was cytotoxic for rectal cancer. The authors observed no significant difference in cytotoxicity between gold nanoparticles and cetuximab compared with cetuximab alone, but it remains to be seen whether this was due to differences in cancer types [27].

In addition, PDL1-targeted therapies, such as pembrolizumab, have become major treatment options since the KEYNOTE048 trial, and they are currently being combined with cisplatin and other anticancer agents. The possibility of combination therapy with immune checkpoint agents should be considered to achieve higher antitumor efficacy [28,29,30].

Bioradiotherapy (BRT) is an option for treating HNSCC. BRT combines cetuximab with radiotherapy [31]. BRT completion rates and objective response rates were reported to be at 78% and 84%, respectively [31]. Radiotherapy is one of the most important treatments in cancer treatment. AuNPs, which enhance the effects of radiotherapy, are expected to be applied clinically in the future [19,21]. AuNPs alone do not display tumor-suppressing effects, but they have been revealed to increase the effects of cetuximab and irradiation. Increasing the dose of cetuximab is expected to result in a stronger inhibitory effect, but this is also believed to considerably increase side effects.

The limitations of this study included the short duration of observation and the use of only one cell line, and it is unclear whether similar effects would be obtained with other cell lines. Because the model was oral cancer, both AuNPs and cetuximab can be injected directly, but it might be difficult to use this combination regimen for tumors in other locations.

In addition, if the number of injected cells is increased, then the tumor size will increase, and it will be easier to inject AuNPs and cetuximab into the tumor.

## 5. Conclusions

AuNPs and cetuximab can be used in combination to exert antitumor effects on HSC-3 cells. Furthermore, radiotherapy is expected to enhance their effects. Because this experiment was also conducted in vivo, we believe that this strategy will be effective in living organisms.

## Figures and Tables

**Figure 1 cancers-15-05697-f001:**
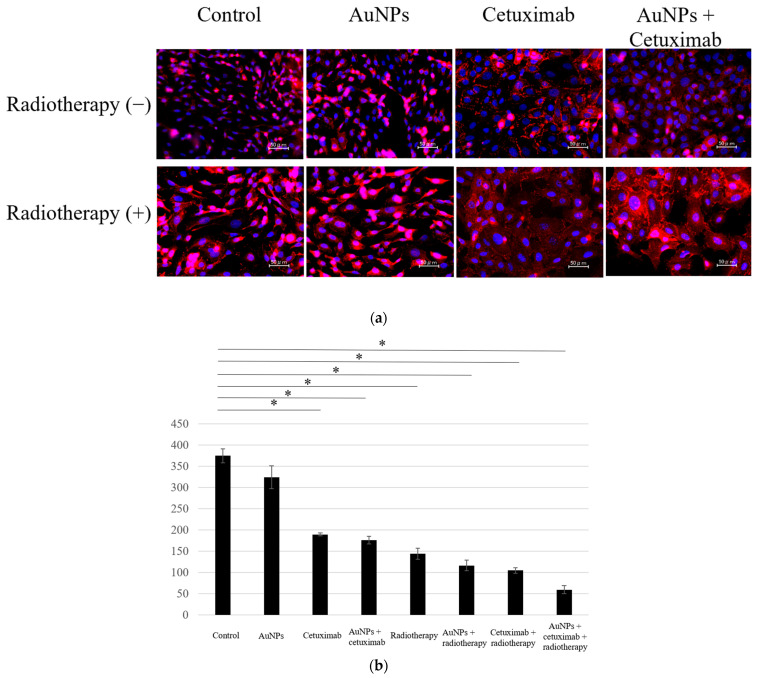
(**a**) Cells were divided into eight groups and immunostained using DAPI and E-caderin, after which the number of cells was counted. (**b**) Compared with the control group, the cell count was lower in all treatment groups, excluding the AuNPs group. The largest decrease was observed in the AuNPs + cetuximab + radiotherapy group. AuNPs, gold nanoparticles. * (Asterisk marks) indicate statistically significant differences.

**Figure 2 cancers-15-05697-f002:**
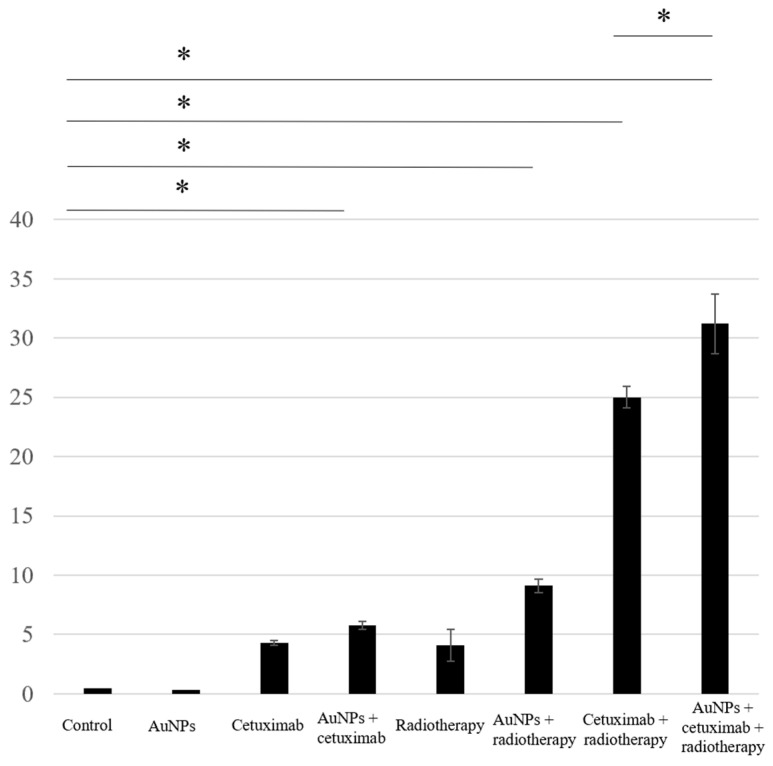
Based on the number of caspase-3-positive cells, the highest rate of apoptosis was observed in the AuNPs + cetuximab + radiotherapy group, followed by the cetuximab + radiotherapy group. AuNPs, gold nanoparticles. * (Asterisk marks) indicate statistically significant differences.

**Figure 3 cancers-15-05697-f003:**
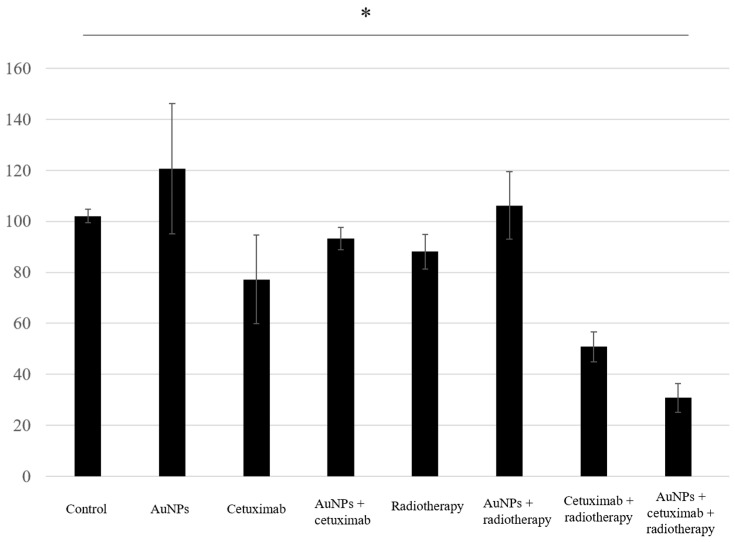
The combination of AuNPs and cetuximab inhibited cell proliferation, and this effect was strengthened by the addition of radiotherapy. The Y-axis presents percentages based on the control as 100%. Data are presented as the mean ± standard deviation of three independent experiments. AuNPs, gold nanoparticles. * (Asterisk marks) indicate statistically significant differences.

**Figure 4 cancers-15-05697-f004:**
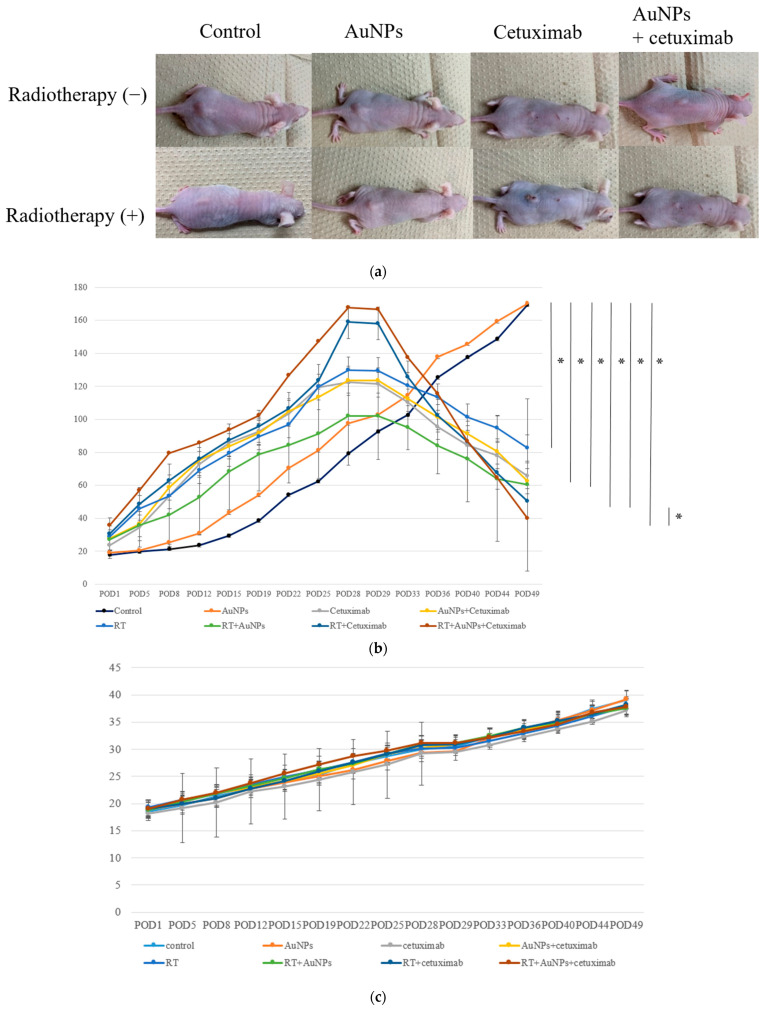
(**a**) Photographs of subcutaneous tumors on the backs of eight groups of representative nude mice at POD49. (**b**) After approximately 4 weeks, the tumor volume reached its maximum, after which point AuNPs and cetuximab were injected into the tumors. On the next day, tumors were irradiated and then monitored for 3 weeks. The tumor volume continued to increase in the control and AuNPs groups, whereas tumor shrinkage was noted in the other groups. The greatest tumor shrinkage was observed in the AuNPs + cetuximab + radiotherapy group. (**c**) Body weight tended to increase in all individuals, and no significant difference was observed among the groups. AuNPs, gold nanoparticles; RT, radiotherapy; POD, postoperative day. * (Asterisk marks) indicate statistically significant differences.

**Figure 5 cancers-15-05697-f005:**
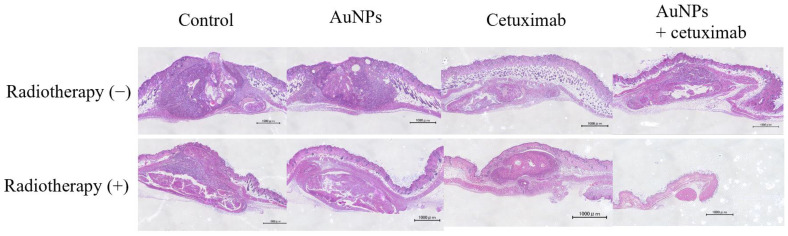
Tumor sections were examined under a microscope. Based on the tumor volume of the cross-section, the greatest shrinkage was observed in the AuNPs + cetuximab + radiotherapy group. Scale bars: 1000 μm. AuNPs, gold nanoparticles.

**Figure 6 cancers-15-05697-f006:**
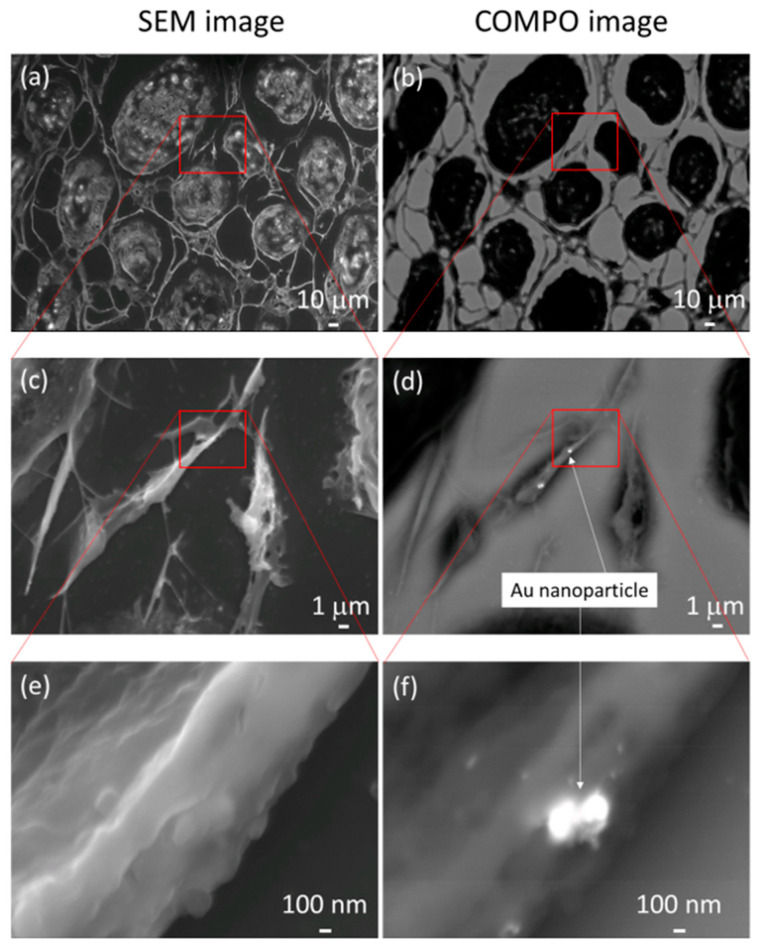
Panels (**a**,**c**,**e**) present normal scanning electron microscopy images. The corresponding backscattered electron compositional (COMPO) images are presented in panels (**b**,**d**,**f**). In the COMPO images, the white areas are the locations in which the reflected electron emission rate is high, i.e., where atoms heavier than carbon, the main component of the cell, are present. Gold nanoparticles that are hidden by part of the cytoskeleton in (**c**,**e**) can be clearly observed in (**d**,**f**).

**Figure 7 cancers-15-05697-f007:**
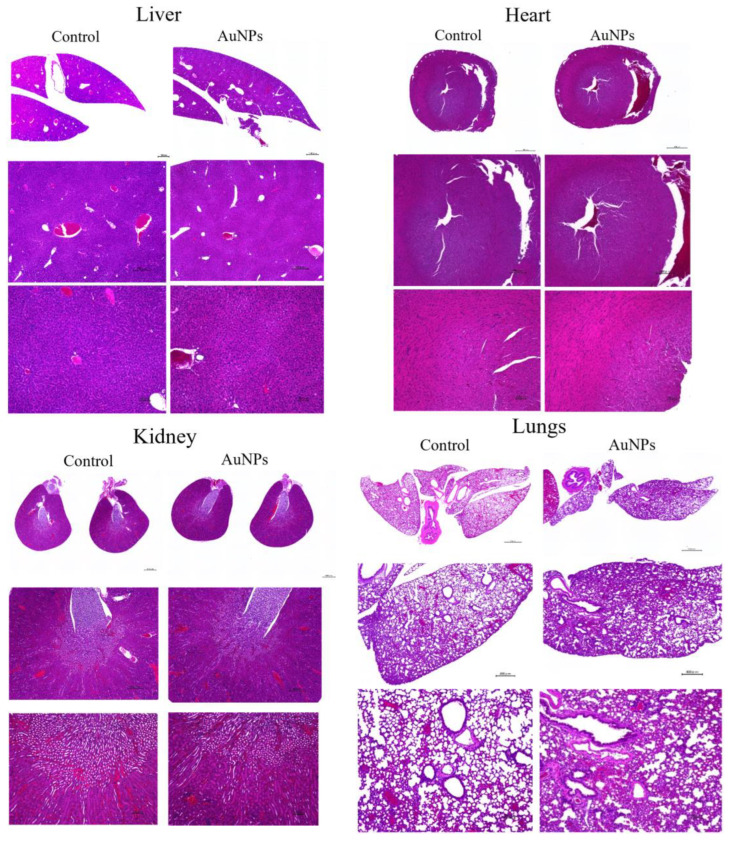
To assess the toxicity of AuNPs in organs, major organs such as the liver, heart, kidney, and lung were removed from mice in the control and AuNPs groups and stained with hematoxylin and eosin. No obvious toxicity findings were observed. Scale bars: top panel, 1000 μm; middle panel, 400 μm; bottom panel, 100 μm. AuNPs, gold nanoparticles.

## Data Availability

All data generated or analyzed during this study are included in this published article. The data that support the findings of this study are available from the corresponding author upon reasonable request.

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
