# Peer review of "Gold Nanoparticles Enhance the Tumor Growth-Suppressing Effects of Cetuximab and Radiotherapy in Head and Neck Cancer In Vitro and In Vivo"

_cancers, 2023, doi:10.3390/cancers15235697_

Round 1
Reviewer 1 Report
Comments and Suggestions for Authors
This manuscript present potentially interesting results and conclusions
Comments:
1. Abstract: please arrange the abstract in paragraphs headings: Introduction, methods, Results, and Conclusions
2. Results:
For all the quantitative results, including Cell counting Assays, Apoptosis Assay, Proliferation Assay, Tumor Growth Inhibitory Effect, Evaluation of Tumor Cross secrions in vivo, Observation of AuNP uptake, and Toxicity assessment: Please provide in the text the count the numbers in each group and comparisons, as well as the p value, otherwise we cannot assess your statements that the differences were statistically significant.
Author Response
Thank you very much for your peer review.
Responses to the comments of Reviewer 1
Comment 1- Abstract
please arrange the abstract in paragraphs headings: Introduction, methods, Results, and Conclusions..
Response: Thank you for your suggestion. Subheadings have been added to the abstract.
“Introduction: Head and neck squamous cell carcinoma (HNSCC) treatment includes surgery, radiotherapy, and immunotherapy with the aim of eradicating cancer cells without affecting normal tissues. HNSCC expresses epidermal growth factor receptor (EGFR), and cetuximab, an IgG1 monoclonal antibody targeting EGFR, has been approved for the treatment of HNSCC. However, cetuximab has low reactivity and induces serious side effects. Gold nanoparticles (AuNPs) were reported to enhance the local antitumor effects of radiotherapy without damaging normal cells. Methods and Results: This study investigated the in vitro effects of single and combination therapy with AuNPs (1.0 nM), cetuximab (30 nM), and radiotherapy (4 Gy) on a human HNSCC cell line, HSC-3. Combination treatment of AuNPs+cetuximab+radiotherapy markedly reduced HSC-3 numbers and proliferation, and enhanced apoptosis compared with single and double combination treatments. Furthermore, the in vivo combination treatment (AuNPs+cetuximab+radiotherapy) of a xenograft model of HSC-3 cells transplanted into nude mice (BALB/cAJcl-nu/nu) reduced the tumor volume compared with controls. Scanning electron microscopy demonstrated the presence of AuNPs in tumor tissues and toxicity analysis indicated AuNPs had no toxic effect on normal tissues. Conclusions: This study showed that AuNPs alone do not have a tumor-suppressing effect, but they sensitize tumors to radiotherapy and bind to cetuximab, leading to enhanced antitumor effects.”
Comment 2- Results
“For all the quantitative results, including Cell counting Assays, Apoptosis Assay, Proliferation Assay, Tumor Growth Inhibitory Effect, Evaluation of Tumor Cross secrions in vivo, Observation of AuNP uptake, and Toxicity assessment: Please provide in the text the count the numbers in each group and comparisons, as well as the p value, otherwise we cannot assess your statements that the differences were statistically significant..” ïƒ
Response: To address the reviewer’s recommendation, we changed the first to the fourth paragraph of the Results to the following:
“• Cell Counting Assay
We determined the average counts of HSC-3 cells in the eight different experimental groups. Cell counting assay results showed that the average number of cells in each group was 374.4 for control, 324.1 for AuNPs, 189.6 for cetuximab, and 176.7 for AuNPs plus cetuximab. When radiation was added to each condition, the average number of cells was 148.6, 122.7, 100.2, and 59.5, respectively. The cell count was significantly reduced in all treatment groups versus the control group (p<0.001), excluding the AuNPs alone group, as presented in Figure 1.
- Apoptosis Assay
We assessed apoptosis in HSC-3 cells following treatment with AuNPs, cetuximab, and/or radiotherapy by counting the number of cells positively stained for caspase-3. The average percentages of apoptosis-positive cells for control, AuNPs, cetuximab, and AuNPs plus cetuximab were 0.4%, 0.3%, 4.3%, and 5.8%, respectively, and increased to 4.1%, 9.1%, 25%, and 31% when radiation was added to each treatment. Significant differences were observed between the control and AuNPs + radiation groups, the control and radiation + cetuximab groups, and the control and radiation + cetuximab + AuNPs groups (p<0.001). Significant differences were also observed between the radiation + cetuximab and radiation + cetuximab + AuNPs groups (p = 0.0144). Significantly more apoptotic cells were detected in the AuNPs + cetuximab, AuNPs + radiotherapy, cetuximab + radiotherapy, and AuNPs + cetuximab + radiotherapy groups than in the control group. In addition, a significantly greater percentage of cells were apoptotic in the AuNPs + cetuximab + radiotherapy group than in the cetuximab + radiotherapy group. The detailed results are presented in Figure 2
- Proliferation Assay
Proliferation assay results showed that the average cell growth percentages were 102% for control, 120% for AuNPs, 77% for cetuximab, and 93% for AuNPs plus cetuximab, and 88%, 106%, 51%, and 31% when radiation was added to each condition. Cell proliferation was significantly reduced in the AuNPs + cetuximab + radiotherapy group compared with that in the control group (p=0.0049) The detailed results are presented in Figure 3.
- Tumor Growth-Inhibitory Effect In Vivo
The tumor size in mice injected with HSC-3 cells was approximately 140–160 mm3 prior to treatment.
As shown in Figure 4(a), at POD49, the subcutaneous tumor volume had decreased in the treatment groups versus the control and AuNPs only groups. The average volume of the control group was 169.3 mm3, the volume of AuNPs was 170.2 mm3, cetuximab was 65.7 mm3, and AuNPs plus cetuximab was 62.4 mm3. Adding radiation to each group resulted in tumor volumes of 82.6 mm3, 60.3 mm3, 50.2 mm3 and 39.8 mm3. On day 49 after treatment, the tumor volume was significantly lower in all treatment groups, excluding the AuNPs group, than in the control group (Figure 4(b)). In addition, tumor volume was significantly smaller in the AuNPs + cetuximab + radiotherapy group than in the cetuximab + radiotherapy group (p = 0.0036). Body weight tended to increase in all animals, and no significant difference was observed among the groups. (Figure 4(c)).”
Reviewer 2 Report
Comments and Suggestions for Authors
This manuscript describes the combined method of chemotherapy and radiotherapy for HNSCC treatment. The idea seems to be novel. However, this manuscript has a bad content organization, scientific result discussion, method introduction, etc. Therefore, I would strongly recommend that this manuscript is not ready to be published unless the authors can solve and nicely answer my questions below.
1. In the introduction part, the authors introduced the gold nanoparticles and very briefly introduced the gold nanoparticle properties, which I think is not sufficient. The author mentioned the (localized) surface plasma resonance (LSPR) of gold nanoparticles. Is this related to the cancer treatment? For 10nm gold nanoparticles, the LSPR bandgap is around 515 nm. In the manuscript, there is nothing related to the application of a 515nm green laser. Please elaborate on the potential role that the gold nanoparticle plays in your experiment.
2. Most importantly, please provide the source of the gold nanoparticles. Did you synthesize the gold nanoparticles? What is the synthesis method? What is the surface coating? If possible, please collect TEM or DLS to confirm the size of the particle.
3. In vitro study, does gold nanoparticle exist inside the cell? If so, please provide TEM to confirm.
4. Also very importantly, what is the injection method of gold nanoparticle and cetuximab during the in vivo study? Local injection, or intravenous injection?
5. What is the dosage information during the in vivo treatment? Please provide detailed information including the volume and the concentration of each ingredient.
6. The figures have really bad caption. There are multiple sections in each figure, but each section is not clearly described individually and separately in the caption.
7. Why there is a literature review in the Discussion section (lines 299 to 336)? This section should be included in the Introduction section.
8. The Result and Discussion section is really badly written. The authors simply introduce each figure WITHOUT detailed discussion scientifically.
9. Overall, I don't understand the purpose of the gold nanoparticle addition. What is the role that gold nanoparticles play during the experiment? From my perspective, it is related to the radiotherapy under the X-ray. If so, the authors should provide detailed information to support this. If not, the authors should clarify this.
Author Response
Thank you very much for your peer review.
Responses to the comments of Reviewer 2
Comment
This manuscript describes the combined method of chemotherapy and radiotherapy for HNSCC treatment. The idea seems to be novel. However, this manuscript has a bad content organization, scientific result discussion, method introduction, etc. Therefore, I would strongly recommend that this manuscript is not ready to be published unless the authors can solve and nicely answer my questions below.
Response: Thank you for pointing this out. We have made changes to the manuscript in response to each of your suggestions.
Comment 1- Introduction
“In the introduction part, the authors introduced the gold nanoparticles and very briefly introduced the gold nanoparticle properties, which I think is not sufficient. The author mentioned the (localized) surface plasma resonance (LSPR) of gold nanoparticles. Is this related to the cancer treatment? For 10nm gold nanoparticles, the LSPR bandgap is around 515 nm. In the manuscript, there is nothing related to the application of a 515nm green laser. Please elaborate on the potential role that the gold nanoparticle plays in your experiment”.
Response: An ideal nanoparticle for diagnosis and therapy using photothermal ablation should have the following features: (1) a large absorption cross-section for the excited light, especially near infrared light, (2) a size below 100 nm to enhance tumor uptake and to reduce sequestration, and (3) low toxicity and biocompatibility. AuNPs are expected to be the most suitable nanoparticle for diagnosis and therapy because of their biocompatibility and low cytotoxicity. [1, 2]
According to previous studies [1 – 7], AuNPs have immense potential for cancer diagnosis and therapy on account of their surface plasmon resonance resulting from enhancement of light scattering and absorption. [1 – 7] Nanoparticle-mediated photothermal ablation of tumors can provide an emerging tool for a novel therapy against cancer. For example, Ayala-Orozco et al. demonstrated the photothermal ablation of cancerous tumors following nanoparticle uptake at the tumor site. This study revealed that their fabricated ~90 nm diameter Au nanoparticles (Au/SiO2/Su), which are strong light absorbers with 77 % absorption efficiency, are more effective than Au nanoshells for photothermal cancer therapy in tumor-bearing mice. [5] However, conjugation of AuNPs to ligands specifically targeted to biomarkers on cancer cells enables molecular-specific imaging and detection of cancer. [4] Thus, the near-infrared absorbing Au-based nanoparticles serving as photothermal transducers are the key components for cancer therapy and diagnosis.
In the present study, the therapeutic effects of anticancer drugs combined with irradiation was investigated instead of photothermal therapy with AuNPs. AuNPs used in photothermal therapy have a plasmonic structure and are usually approximately 50 nm to 200 nm in size and are not internalized into cells; nanoparticles smaller than approximately 20 nm are taken endocytosed into cells and are likely to increase toxicity. However, gold is less cytotoxic and is therefore considered suitable for the purpose of the present study. In previous studies [6, 7], AuNPs were not demonstrated to be taken up by cells and were localized at the receptor area on the cell surface. To enhance the therapeutic effect, the current study will be conducted to evaluate the effect of AuNPs that are internalized by cancer cells as a result of their reduced size.
To clarify this question in the manuscript text, we have added the following statements in the Introduction: “As a result of their high atomic number, gold nanoparticles are ideal radiosensitizers that absorb photons and emit secondary photoelectrons, which may enhance the cell-killing properties of radiation through DNA damage [16].”
[1] A. Sani, C. Cao, and D. Gui, Biochemistry and Biophysics Reports, 26, 100991 (2021).
[2] C. Carnovale, G. Bryant, R. Shukla, and V. Bansal, ASC OMEGA, 4, 242 (2019).
[3] S. Lal, S. E. Clare, and N. J. Halas, Acc. Chem. Res. 41, 12, 1842–1851 (2008).
[4] P. K. Jain, I. H. El-Sayed, M. A. El-Sayed, Nanotoday 2, 18 – 29 (2007).
[5] C. Ayala-Orozco, C. Urban, M. W. Knight, A. S. Urban, O. Neumann, S. W. Bishnoi, S. Mukherjee, A. M. Goodman, H. Charron, T. Mitchell, M. Shea, R. Roy, S. Nanda, R. Schiff⊥, N. J. Halas, and A. Joshi, ACS Nano , 8, 6, 6372–6381 (2014).
[6] Kashin, M.; Kakei, Y.; Teraoka, S.; Hasegawa, T.; Yamaguchi, A.; Fukuoka, T.; Sasaki, R.; Akashi, M. Gold Nanoparticles Enhance EGFR Inhibition and Irradiation Effects in Head and Neck Squamous Carcinoma Cells. Biomed Res Int. 2020, 7, 1281645.
[7] Teraoka, S.; Kakei, Y.; Akashi, M.; Iwata, E.; Hasegawa, T.; Miyawaki, D.; Sasaki, R.; Komori, T. Gold nanoparticles enhance X ray irradiation induced apoptosis in head and neck squamous cell carcinoma in vitro. Biomed Rep. 2018, 9, 415-420
Comment 2- Method
“Most importantly, please provide the source of the gold nanoparticles. Did you synthesize the gold nanoparticles? What is the synthesis method? What is the surface coating? If possible, please collect TEM or DLS to confirm the size of the particle.”
Response: In both in vitro and in vivo studies, gold nanoparticles were purchased from the manufacturer as indicated in the revised text. To address the reviewer’s recommendation, we added additional details to the Methods section as follows:
“Citrate-stabilized AuNPs were purchased from Cytodiagnostics, Inc. (Burlington, ON, Canada). Standard AuNPs (60 nm) were used (lot number: 2458052_60). Lysosomal uptake of AuNPs was confirmed by transmission electron microscope (JEM-1400Plus; JEOL Ltd., Tokyo,Japan) imaging at an acceleration voltage of 100 kV. Digital images (3296 × 2472 pixels) were taken with a CCD camera (EM-14830RUBY2; JEOL Ltd.) as reported previously [18] (Supplementary Figure1).”
(5) Xenograft Assay
HSC-3 cells (3.5 × 106–4.0 × 106 cells) were mixed with Matrigel (Basement Membrane Matrix, Corning Inc., Shizuoka, Japan) and injected at a volume of 0.1 mL subcutaneously into the back of each nude mouse. After the tumor volume reached 200–300 mm3, mice were assigned to one of eight groups: control (treated with PBS), AuNPs (Product Number 765309, Sigma-Aldrich; Merck KGaA, Darmstadt, Germany) (treated with 10 nm diameter AuNP suspension at a concentration of 15 μg mL−1), cetuximab (treated with cetuximab suspension at a concentration of 500 μg mL−1), AuNPs + cetuximab (treated with 10 nm diameter AuNP suspension at a concentration of 15 μg mL−1 and cetuximab suspension at a concentration of 500 μg mL−1), radiotherapy (4 Gy), AuNP + radiotherapy (treated with 10 nm diameter AuNP suspension at a concentration of 15 μg mL−1 and 4 Gy or radiation), cetuximab + radiotherapy (treated with cetuximab suspension at a concentration of 500 μg mL−1 and 4 Gy of radiation), and AuNP + cetuximab + radiotherapy (treated with 10 nm diameter AuNP suspension at a concentration of 15 μg mL−1, cetuximab suspension at a concentration of 500 μg mL−1, and 4 Gy of radiation).
Comment 3- Results
“In vitro study, does gold nanoparticle exist inside the cell? If so, please provide TEM to confirm.…”
Response: Yes, we have confirmed by TEM as previously reported that gold nanoparticles are present internally in cells in in vitro studies. We have added additional details and a Supplementary Figure to clarify:
“Citrate-stabilized AuNPs were purchased from Cytodiagnostics, Inc. (Burlington, ON, Canada). Standard AuNPs (60 nm) were used (lot number: 2458052_60). Lysosomal uptake of AuNPs was confirmed by transmission electron microscope (JEM-1400Plus; JEOL Ltd., Tokyo,Japan) imaging at an acceleration voltage of 100 kV. Digital images (3296 × 2472 pixels) were taken with a CCD camera (EM-14830RUBY2; JEOL Ltd.) as reported previously [18] (Supplementary Figure1).”
“Comment 4- Methods
“Also very importantly, what is the injection method of gold nanoparticle and cetuximab during the in vivo study? Local injection, or intravenous injection?” ïƒ
Response: We thank the reviewer for this comment. In response, we made the following change to the manuscript:
“HSC-3 cells (3.5 × 106–4.0 × 106 cells) were mixed with Matrigel (Basement Membrane Matrix, Corning Inc., Shizuoka, Japan) and injected at a volume of 0.1 mL subcutaneously into the back of each nude mouse. After the tumor volume reached 200–300 mm3, mice were assigned to one of eight groups: control (treated with PBS), AuNPs (Product Number 765309, Sigma-Aldrich; Merck KGaA, Darmstadt, Germany) (treated with 10 nm diameter AuNP suspension at a concentration of 15 μg mL−1), cetuximab (treated with cetuximab suspension at a concentration of 500 μg mL−1), AuNPs + cetuximab (treated with 10 nm diameter AuNP suspension at a concentration of 15 μg mL−1 and cetuximab suspension at a concentration of 500 μg mL−1), radiotherapy (4 Gy), AuNP + radiotherapy (treated with 10 nm diameter AuNP suspension at a concentration of 15 μg mL−1 and 4 Gy or radiation), cetuximab + radiotherapy (treated with cetuximab suspension at a concentration of 500 μg mL−1 and 4 Gy of radiation), and AuNP + cetuximab + radiotherapy (treated with 10 nm diameter AuNP suspension at a concentration of 15 μg mL−1, cetuximab suspension at a concentration of 500 μg mL−1, and 4 Gy of radiation). AuNPs, cetuximab, or the combination of both were adjusted to a total of 150 μl in cell culture medium to the final concentrations described above, and then injected directly into the subcutaneous tumor on the backs of nude mice. X-ray irradiation was performed using the MBR-1505R2 X-ray generator (Hitachi, Tokyo, Japan) at a voltage of 150 kV and a current of 5 mA with a 1-mm thick aluminum filter (0.5 Gy min−1 at the target).”
Comment 5- INTRODUCTION
“What is the dosage information during the in vivo treatment? Please provide detailed information including the volume and the concentration of each ingredient.” ïƒ
Response: To address the reviewers’ recommendation, we revised the Methods to the following:
“HSC-3 cells (3.5 × 106–4.0 × 106 cells) were mixed with Matrigel (Basement Membrane Matrix, Corning Inc., Shizuoka, Japan) and injected at a volume of 0.1 mL subcutaneously into the back of each nude mouse. After the tumor volume reached 200–300 mm3, mice were assigned to one of eight groups: control (treated with PBS), AuNPs (Product Number 765309, Sigma-Aldrich; Merck KGaA, Darmstadt, Germany) (treated with 10 nm diameter AuNP suspension at a concentration of 15 μg mL−1), cetuximab (treated with cetuximab suspension at a concentration of 500 μg mL−1), AuNPs + cetuximab (treated with 10 nm diameter AuNP suspension at a concentration of 15 μg mL−1 and cetuximab suspension at a concentration of 500 μg mL−1), radiotherapy (4 Gy), AuNP + radiotherapy (treated with 10 nm diameter AuNP suspension at a concentration of 15 μg mL−1 and 4 Gy or radiation), cetuximab + radiotherapy (treated with cetuximab suspension at a concentration of 500 μg mL−1 and 4 Gy of radiation), and AuNP + cetuximab + radiotherapy (treated with 10 nm diameter AuNP suspension at a concentration of 15 μg mL−1, cetuximab suspension at a concentration of 500 μg mL−1, and 4 Gy of radiation). AuNPs, cetuximab, or the combination of both were adjusted to a total of 150 μl in cell culture medium to the final concentrations described above, and then injected directly into the subcutaneous tumor on the backs of nude mice. X-ray irradiation was performed using the MBR-1505R2 X-ray generator (Hitachi, Tokyo, Japan) at a voltage of 150 kV and a current of 5 mA with a 1-mm thick aluminum filter (0.5 Gy min−1 at the target).”
Comment 6- Results
“The figures have really bad caption. There are multiple sections in each figure, but each section is not clearly described individually and separately in the caption.”
.
Response: To address the reviewer’s recommendation, we have changed the captions to the following:
“Figure 4. (a) Photographs of subcutaneous tumors on the backs of eight groups of representative nude mice at POD49. (b) After approximately 4 weeks, the tumor volume reached its maximum, after which point AuNPs and cetuximab were injected into the tumors. On the next day, tumors were irradiated and then monitored for 3 weeks. The tumor volume continued to increase in the control and AuNPs groups, whereas tumor shrinkage was noted in the other groups. The greatest tumor shrinkage was observed in the AuNPs + cetuximab + radiotherapy group. (c) Body weight tended to increase in all individuals, and no significant difference was observed among the groups. AuNPs, gold nanoparticles; RT, radiotherapy; POD, postoperative day
Figure 7. To assess the toxicity of AuNPs in organs, various major organs such as liver, heart, kidney, and lung were removed from mice in the control and AuNPs groups and stained with hematoxylin and eosin. No obvious toxicity findings were observed. Scale bars: top panel, 1000 μm; middle panel, 400 μm; bottom panel, 100 μm.”
AuNPs, gold nanoparticles.”
Comment 7- Discussion
- “Why there is a literature review in the Discussion section (lines 299 to 336)? This section should be included in the Introduction section.”
Response: To address the reviewers’ recommendation, we changed the first and second paragraph of the Discussion to provide additional commentary on how the present study aligns with previous work:
“This study revealed that combination treatment with AuNPs, cetuximab, and radiotherapy significantly reduced HSC-3 cell counts (Figure 1, 2, and 3.). Furthermore, this combination treatment led to tumor shrinkage in vivo as shown in Figure 4, 5 without toxic findings in major organs, as shown in Figure7. As shown in Figure 6, the strength of this study also lies in the demonstration of the localization of gold nanoparticles within the excised tumor. AuNPs alone do not have a tumor-suppressing effect, but they sensitize tumors to radiotherapy and bind to cetuximab, leading to enhanced antitumor effects. (Figure 1, 2, 3, 4, and 5) The reduction in cell counts following treatment with AuNPs, cetuximab, and radiotherapy was attributable to apoptosis (Figure2). This is the first report to examine the combined effects of AuNPs, cetuximab, and radiotherapy in mice with HNSCC.
Hassan et al. compared the radiosensitizing effects of TiOxNPs and AuNPs in MIA PaCa-2 human pancreatic cancer cells (JCRB0070) and found that TiOxNPs enhanced tumor suppression [20]. This study compared gold nanoparticles with titanium nanoparticles but did not add molecularly targeted drugs such as cetuximab, such as was done in the current study. In other reports, cetuximab and trastuzumab were used to direct AuNPs toward cancer to enhance the effects of radiotherapy [22-27]. Chattopadhyay et al. reported that HER2-targeted AuNPs increased the antitumor effect of radiotherapy in a xenograft model of breast cancer and that AuNPs are not harmful to normal tissue in vitro and in vivo [22, 23] Yook et al. found that AuNPs targeting both HER2 and EGFR had a stronger radiosensitizing effect on breast cancer cell lines than AuNPs targeting each gene alone [24] The methodology used in these studies was similar to ours, but differed in that breast cancer was the subject of the studies. Popovtzer et al. reported that cetuximab targeted with AuNPs for head and neck cancer enhanced the effects of radiotherapy and significantly affected tumor growth, and the mechanism of radiation enhancement was associated with increased apoptosis (TUNEL assay), inhibition of angiogenesis (based on CD34 levels), and decreased repair mechanisms (proliferating cell nuclear antigen staining). Furthermore, they reported that AuNPs were safe because no evidence of toxicity was observed [25]. This study is the only in vivo report using gold nanoparticles, radiation, and cetuximab; however, the cell line used for head and neck cancer was epidermoid carcinoma, a rare histology for human head and neck squamous cell carcinoma, and the authors did not identify the gold nanoparticles in the tumor tissue. The addition of AuNPs in this study increased the antitumor effects of radiotherapy and cetuximab, but the combination did not completely eliminate tumors in vivo. Cetuximab has been clinically applied in combination with taxane-based anticancer agents such as paclitaxel in the treatment of difficult-to-resect tumors [26]. Hallal et al. reported that the fixed-dose combination of gold nanoparticles and cetuximab itself was cytotoxic for rectal cancer. The authors observed no significant difference in cytotoxicity between gold nanoparticles and cetuximab compared with cetuximab alone, but it remains to be seen whether this was due to differences in cancer types.[27].”
Comment 8- Result and Discussion section: -
“The Result and Discussion section is really badly written. The authors simply introduce each figure WITHOUT detailed discussion scientifically.”
Response: In response, we made the following changes to Results and Discussion:
“• Cell Counting Assay
We determined the average counts of HSC-3 cells in eight different groups. Cell counting assay results showed that the average number of cells in each group was 374.4 for control, 324.1 for AuNPs, 189.6 for cetuximab, and 176.7 for AuNPs plus cetuximab, when radiation was added to each, 148.6 and 122.7, 100.2 and 59.5, respectively. The cell count was significantly reduced in all treatment groups, excluding the AuNPs group, versus the control group (p<0.001), as presented in Figure 1.
- Apoptosis Assay
We assessed apoptosis in HSC-3 cells following treatment with AuNPs, cetuximab, and/or radiotherapy by counting the number of cells positively stained for caspase-3. The average percentages of apoptosis-positive cells for control, AuNPs, cetuximab, and AuNPs plus cetuximab were 0.4%, 0.3%, 4.3%, and 5.8%, respectively, and increased to 4.1%, 9.1%, 25%, and 31% when radiation was added to each treatment, respectively. Significant differences were observed between the control and AuNPs + radiation groups, the control and radiation + cetuximab groups, and the control and radiation + cetuximab + AuNPs, respectively (p<0.001). Significant differences were also observed between the radiation + cetuximab and radiation + cetuximab + AuNPs groups (p = 0.0144).Significantly more apoptotic cells were detected in the AuNPs + cetuximab, AuNPs + radiotherapy, cetuximab + radiotherapy, and AuNPs + cetuximab + radiotherapy groups than in the control group. In addition, a significantly greater percentage of cells were apoptotic in the AuNPs + cetuximab + radiotherapy group than in the cetuximab + radiotherapy group. The detailed results are presented in Figure 2.
- Proliferation Assay
Proliferation assay results showed that the respective average percentages were 102% for control, 120% for AuNPs, 77% for cetuximab, and 93% for AuNPs plus cetuximab, and 88%, 106%, 51%, and 31% for radiation added to each. Cell proliferation was significantly reduced in the AuNPs + cetuximab + radiotherapy group compared with the findings in the control group (p=0.0049) The detailed results are presented in Figure 3. (Figure 3).
- Tumor Growth-Inhibitory Effect In Vivo
The tumor size in mice injected with HSC-3 cells was approximately 140–160 mm3 prior to treatment.
As shown in Figure 4(a), at POD49, the subcutaneous tumor volume had decreased in the treatment groups versus the control and AuNPs only groups. The average volume of the control group was 169.3 mm3, the volume of AuNPs was 170.2 mm3, cetuximab was 65.7 mm3, and AuNPs plus cetuximab was 62.4 mm3. Adding radiation to each group resulted in tumor volumes of 82.6 mm3, 60.3 mm3, 50.2 mm3 and 39.8 mm3.On day 49 after treatment, the tumor volume was significantly lower in all treatment groups, excluding the AuNPs group, than in the control group (Figure 4(b)). In addition, tumor volume was significantly smaller in the AuNPs + cetuximab + radiotherapy group than in the cetuximab + radiotherapy group (p = 0.0036). Body weight tended to increase in all animals, and no significant difference was observed among the groups. (Figure 4(c)).
- Discussion
This study revealed that combination treatment with AuNPs, cetuximab, and radiotherapy significantly reduced HSC-3 cell counts (Figure 1, 2, and 3.). Furthermore, this combination treatment led to tumor shrinkage in vivo as shown in Figure 4, 5 without toxic findings in major organs, as shown in Figure 7. As shown in Figure 6, the strength of this study also lies in the demonstration of the localization of gold nanoparticles within the excised tumor. AuNPs alone do not have a tumor-suppressing effect, but they sensitize tumors to radiotherapy and bind to cetuximab, leading to enhanced antitumor effects. (Figure 1, 2, 3, 4, and 5) The reduction in cell counts following treatment with AuNPs, cetuximab, and radiotherapy was attributable to apoptosis (Figure 2). This is the first report to examine the combined effects of AuNPs, cetuximab, and radiotherapy in mice with HNSCC.
Comment 9-
“Overall, I don't understand the purpose of the gold nanoparticle addition. What is the role that gold nanoparticles play during the experiment? From my perspective, it is related to the radiotherapy under the X-ray. If so, the authors should provide detailed information to support this. If not, the authors should clarify this.” ïƒ
Response: We thank the reviewer for this comment. In response, we made the following change to the Introduction section of the manuscript:
“Therefore, gold nanoparticles (AuNPs) have attracted attention because they only enhance the local action of radiation. AuNPs have found wide application in biology and engineering because of their characteristic optical properties [14-15], which are attributable to the interaction of electrons and light on the particles. As a result of their high atomic number, gold nanoparticles are ideal radiosensitizers that absorb photons and emit secondary photoelectrons, which may enhance the cell-killing properties of radiation through DNA damage [16]. At certain wavelengths of light, a phenomenon called surface plasmon resonance occurs. In this process, electrons on the particle surface oscillate together, resulting in strong absorption and scattering of light [17]. In this experiment, cetuximab was used instead of the selective EGFR tyrosine kinase inhibitor AG1478 [18], which was used in previous experiments. Gold nanoparticles are attached to inhibitors such as EGFR, as in our previous study [19]. We propose that if an EGFR inhibitor with gold nanoparticles on its surface is attached to cancer cells, the radiosensitizing effect of the EGFR inhibitor may be enhanced. “
Reviewer 3 Report
Comments and Suggestions for Authors
Gold nanoparticles enhance the tumor growth-suppressing effects of cetuximab and radiotherapy in head and neck cancer in vitro and in vivo.
This is the well-designed study showing use of gold nanoparticles along with cetuximab in combination with radiotherapy to achieve the anti-cancer activity.
1. What is rationale of choosing squamous cell carcinoma as the model of the study and no other cancers type?
2. What is the rationale of selecting HSC-3 cell line only as ATCC head and neck cancer panel has A-253, SCC-15, SCC-25, SCC-9, FaDu and Detroit 562 please justify.
3. line 120 HSC-3 cells (3.5 × 106–4.0 × 106 cells) please make superscript to 106.
4. What was rate dose rate used please mention?
5. Please mention more about X ray dose used and the settings used for animal studies.
6. Authors have used Isoflurane please mention the concentration of isoflurane used.
7. Does use of isoflurane affects the radiation dose absorbed in the procedure?
8. Line 132 please delete the full stop. .Radiothearpy is one of the---
9. Please mention data about gold nanoparticles characterization for reference please refer the below mentioned article published in molecules in year 2021.
Comments on the Quality of English Language/
Author Response
Thank you very much for your peer review.
Responses to the comments of Reviewer 3
Comment 1
“What is rationale of choosing squamous cell carcinoma as the model of the study and no other cancers type?.”
Response: Thank you for your comment. The quoted document is the latest version of the 2019 data published in the registry of head and neck cancers in Japan [1]. According to these data, a squamous cell carcinoma cell line was selected for this study because, as shown on p. 13, approximately 90% of the histological types, including subtypes, are squamous cell carcinoma, which is the main target for treatment of head and neck cancer.
Comment 2- Results
“What is the rationale of selecting HSC-3 cell line only as ATCC head and neck cancer panel has A-253, SCC-15, SCC-25, SCC-9, FaDu and Detroit 562 please justify...” ïƒ
Response: Thank you for pointing this out. Our laboratory purchases cell lines from the RIKEN BRC CELL BANK in Japan (URL:https://cell.brc.riken.jp/en/ )due to the the cost, ease of transportation, and the fact that most of the cell lines are Japanese. The cell lines available are HSC-3, SAS, and Ca9-22, but Ca9-22 is a gingival carcinoma cell line. The SAS and HSC-3 cell lines are more common tongue cancer cell lines. HSC-3, which we have often used in our laboratory, was used in this study:
Comment 3
“line 120 HSC-3 cells (3.5 × 106–4.0 × 106 cells) please make superscript to 106..” ïƒ
Response: To address the reviewer’s recommendation, this has been updated in the manuscript.
Comment 4
“What was rate dose rate used please mention?” ïƒ
Response: To address the reviewers’ recommendation, we changed the fifth paragraph of the Methods to the following:
“HSC-3 cells (3.5 × 106–4.0 × 106 cells) were mixed with Matrigel (Basement Membrane Matrix, Corning Inc., Shizuoka, Japan) and injected at a volume of 0.1 mL subcutaneously into the back of each nude mouse. After the tumor volume reached 200–300 mm3, mice were assigned to one of eight groups: control (treated with PBS), AuNPs (Product Number 765309, Sigma-Aldrich; Merck KGaA, Darmstadt, Germany) (treated with 10 nm diameter AuNP suspension at a concentration of 15 μg mL−1), cetuximab (treated with cetuximab suspension at a concentration of 500 μg mL−1), AuNPs + cetuximab (treated with 10 nm diameter AuNP suspension at a concentration of 15 μg mL−1 and cetuximab suspension at a concentration of 500 μg mL−1), radiotherapy (4 Gy), AuNP + radiotherapy (treated with 10 nm diameter AuNP suspension at a concentration of 15 μg mL−1 and 4 Gy or radiation), cetuximab + radiotherapy (treated with cetuximab suspension at a concentration of 500 μg mL−1 and 4 Gy of radiation), and AuNP + cetuximab + radiotherapy (treated with 10 nm diameter AuNP suspension at a concentration of 15 μg mL−1, cetuximab suspension at a concentration of 500 μg mL−1, and 4 Gy of radiation). AuNPs, cetuximab, or the combination of both were adjusted to a total of 150 μl in cell culture medium to the final concentrations described above, and then injected directly into the subcutaneous tumor on the backs of nude mice. X-ray irradiation was performed using the MBR-1505R2 X-ray generator (Hitachi, Tokyo, Japan) at a voltage of 150 kV and a current of 5 mA with a 1-mm thick aluminum filter (0.5 Gy min−1 at the target).”
Comment 5
“Please mention more about X ray dose used and the settings used for animal studies.” ïƒ
Response: To address the reviewer’s recommendations, we revised the Methods section as follows:
“HSC-3 cells (3.5 × 106–4.0 × 106 cells) were mixed with Matrigel (Basement Membrane Matrix, Corning Inc., Shizuoka, Japan) and injected at a volume of 0.1 mL subcutaneously into the back of each nude mouse. After the tumor volume reached 200–300 mm3, mice were assigned to one of eight groups: control (treated with PBS), AuNPs (Product Number 765309, Sigma-Aldrich; Merck KGaA, Darmstadt, Germany) (treated with 10 nm diameter AuNP suspension at a concentration of 15 μg mL−1), cetuximab (treated with cetuximab suspension at a concentration of 500 μg mL−1), AuNPs + cetuximab (treated with 10 nm diameter AuNP suspension at a concentration of 15 μg mL−1 and cetuximab suspension at a concentration of 500 μg mL−1), radiotherapy (4 Gy), AuNP + radiotherapy (treated with 10 nm diameter AuNP suspension at a concentration of 15 μg mL−1 and 4 Gy or radiation), cetuximab + radiotherapy (treated with cetuximab suspension at a concentration of 500 μg mL−1 and 4 Gy of radiation), and AuNP + cetuximab + radiotherapy (treated with 10 nm diameter AuNP suspension at a concentration of 15 μg mL−1, cetuximab suspension at a concentration of 500 μg mL−1, and 4 Gy of radiation). AuNPs, cetuximab, or the combination of both were adjusted to a total of 150 μl in cell culture medium to the final concentrations described above, and then injected directly into the subcutaneous tumor on the backs of nude mice. X-ray irradiation was performed using the MBR-1505R2 X-ray generator (Hitachi, Tokyo, Japan) at a voltage of 150 kV and a current of 5 mA with a 1-mm thick aluminum filter (0.5 Gy min−1 at the target).”
Comment 6-
“Authors have used Isoflurane please mention the concentration of isoflurane used..” ïƒ
Response: To address the reviewer’s recommendations, we changed the fifth paragraph of the Methods to the following:
“Prior to each experiment, the mice were anesthetized using 2 % isoflurane in O2 and taped on a base made by combining two tongue depressors. Other body surfaces were covered with lead plates to ensure that only tumors were irradiated.”
Comment 7-
“Does use of isoflurane affects the radiation dose absorbed in the procedure.” ïƒ
Response: Thank you for your comment. While we did not directly assess the impact of isoflurane on the dose of radiation absorbed, we performed the experiments as previously described by the following studies: [1-4].
[1] Niladri Chattopadhyay • Zhongli Cai •Yongkyu Luke Kwon • Eli Lechtman •
Jean-Philippe Pignol • Raymond M. Reilly, Breast Cancer Res Treat, 137, 81-91 (2013).
[2] CHui Li • Laura Diaz • Daniel Lee • Lei Cui •Xin Liang • Yingsheng Cheng, Radiol med, 119, 269-276 (2014).
[3] Linda Karmani, Daniel Labar, Vanessa Valembois, Virginie Bouchat,
Praveen Ganesh Nagaswaran, Anne Bol, Jacques Gillart, Philippe Levêque,
Caroline Bouzin, Davide Bonifazi, Carine Michiels , Olivier Feron,
Vincent Grégoire, Stéphane Lucas, Thierry Vander Borght and
Bernard Gallez, Contrast. Media. Mol. Imaging. 8, 402–408 (2013).
[4] Alexei A. Bogdanov, Jr, Suresh Gupta, Nadezhda Koshkina, Stuart J. Corr, Surong Zhang, Steven A. Curley, and Gang Han, Biocondugate chemistry 26, 39 – 50 (2015).
Comment 8-
“Line 132 please delete the full stop. .Radiothearpy is one of the---.” ïƒ
Response: Thank you for your comment. To address the reviewer’s recommendation, we changed the fifth paragraph of the Discussion to the following:
“Radiotherapy is one of the most important treatments in cancer treatment. AuNPs, which enhance the effects of radiotherapy, are expected to be applied clinically in the future [19, 21]. AuNPs alone do not display tumor-suppressing effects alone, but they have been revealed to increase the effects of cetuximab and irradiation. Increasing the dose of cetuximab is expected to result in a stronger inhibitory effect, but this is also believed to considerably increase side effects.“
Comment 9-
“Please mention data about gold nanoparticles characterization for reference please refer the below mentioned article published in molecules in year 2021..” ïƒ
Response: To address the reviewer’s recommendation, we revised the Discussion to the following:
“Hassan et al. compared the radiosensitizing effects of TiOxNPs and AuNPs in MIA PaCa-2 human pancreatic cancer cells (JCRB0070) and found that TiOxNPs enhanced tumor suppression [20]. This study compared gold nanoparticles with titanium nanoparticles but did not add molecularly targeted drugs such as cetuximab, such as was done in the current study. In other reports, cetuximab and trastuzumab were used to direct AuNPs toward cancer to enhance the effects of radiotherapy [22-27]. Chattopadhyay et al. reported that HER2-targeted AuNPs increased the antitumor effect of radiotherapy in a xenograft model of breast cancer and that AuNPs are not harmful to normal tissue in vitro and in vivo [22, 23] Yook et al. found that AuNPs targeting both HER2 and EGFR had a stronger radiosensitizing effect on breast cancer cell lines than AuNPs targeting each gene alone [24] The methodology used in these studies was similar to ours, but differed in that breast cancer was the subject of the studies. Popovtzer et al. reported that cetuximab targeted with AuNPs for head and neck cancer enhanced the effects of radiotherapy and significantly affected tumor growth, and the mechanism of radiation enhancement was associated with increased apoptosis (TUNEL assay), inhibition of angiogenesis (based on CD34 levels), and decreased repair mechanisms (proliferating cell nuclear antigen staining). Furthermore, they reported that AuNPs were safe because no evidence of toxicity was observed [25]. This study is the only in vivo report using gold nanoparticles, radiation, and cetuximab; however, the cell line used for head and neck cancer was epidermoid carcinoma, a rare histology for human head and neck squamous cell carcinoma, and the authors did not identify the gold nanoparticles in the tumor tissue. The addition of AuNPs in this study increased the antitumor effects of radiotherapy and cetuximab, but the combination did not completely eliminate tumors in vivo. Cetuximab has been clinically applied in combination with taxane-based anticancer agents such as paclitaxel in the treatment of difficult-to-resect tumors [26]. Hallal et al. reported that the fixed-dose combination of gold nanoparticles and cetuximab itself was cytotoxic for rectal cancer. The authors observed no significant difference in cytotoxicity between gold nanoparticles and cetuximab compared with cetuximab alone, but it remains to be seen whether this was due to differences in cancer types.[27].”
Round 2
Reviewer 2 Report
Comments and Suggestions for Authors
My questions and comments are well explained. Therefore, I think this manuscript would be qualified for the publication. Thanks for the well discussion from the authors and researchers.
Reviewer 3 Report
Comments and Suggestions for Authors
Authors have done necessary correction and answered the comments.
Manuscript can be accepted in the present form.